# SWE-PolyBench: A multi-language benchmark for repository level evaluation of coding agents

## Abstract

Coding agents powered by large language models have shown impressive capabilities in software engineering tasks, but evaluating their performance across diverse programming languages and real-world scenarios remains challenging. We introduce SWE-PolyBench, a new multi-language benchmark for repository-level, execution-based evaluation of coding agents. SWE-PolyBench contains 2110 instances from 21 repositories and includes tasks in Java, JavaScript, TypeScript and Python, covering bug fixes, feature additions, and code refactoring. We offer a curated and validated subset of 382 instances (SWE-PolyBench_Verified) featuring high-quality issue descriptions, code and test patches. Additionally, we release an evaluation harness that enables fully automated assessment. We further introduce novel instance stratifications and retrieval metrics rooted in syntax tree analysis to deepen the understanding of coding agent performances. Our experiments with leading open-source coding agents on SWE-PolyBench show that current agents exhibit uneven performances across languages and struggle with complex problems, while showing higher performance on simpler tasks. SWE-PolyBench aims to drive progress in developing more versatile and robust AI coding assistants for real-world software engineering.

## 1 Introduction

Coding agents are autonomous systems based on language models that are able to create or modify software with limited human input. Over the last year, coding agents have garnered substantial attention due to their potential to dramatically enhance human productivity. The current generation of coding agents exhibit impressive performance on a wide-range of text-based tasks like code completion (Guo et al., 2023; Ding et al., 2024), code translation (Szafraniec et al., 2023), documentation generation (Luo et al., 2024), unit test generation (Alshahwan et al., 2024), debugging (Tian et al., 2024), and conversational code generation (Nijkamp et al., 2023). At the same time, their effectiveness in different scenarios is far from being broadly understood. This led to the proliferation of benchmarks aimed at assessing the coding effectiveness of said systems in controlled environments.

In particular, SWE-Bench (Jimenez et al., 2024), which measures the performance of systems at "solving" GitHub issues has spurred the development of capable coding agents resulting in several leaderboard submissions, becoming the de-facto standard for benchmarking a coding agent. Despite its significant impact as a pioneering benchmark, SWE-Bench, and in particular its "verified" subset (Chowdhury et al., 2024), also shows some limitations. It contains only Python repositories, the majority of tasks are bug fixes, and with over 45 % of all tasks, the `Django` repository is significantly over-represented. Lastly, optimizing for a single dataset may result in "overfitted" agents whose capabilities no longer generalize to the broader goal of developing versatile, useful, and robust AI coding assistants.

To address these gaps, we have curated a dataset that aims to provide a diverse benchmarking environment for AI coding agents. Our dataset comprises pull requests (PRs) and issues from 21 repositories from four of the ten most popular languages (according to Stack Overflow (2023)), spanning five categories (feature request, bug fix, refactoring, security, and testing task) and several levels of complexity. Breaking down performance into task categories and complexity can help developers

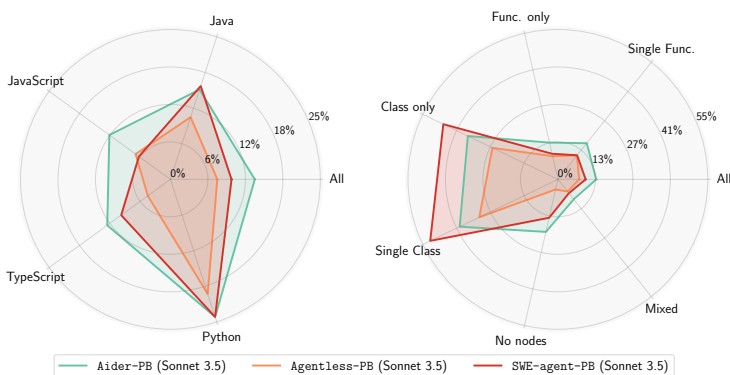

Figure 1: Pass rates of coding agents across programming languages (left) and across subsets of different complexities based on syntax tree analysis (see Section 4 for details on labels).

pinpoint agents' strengths and weaknesses. We further curate an annotated verified subset of 382 instances covering all languages, SWE-PolyBench_Verified, that filters out instances based on issue description, code and test quality.

Our evaluation framework introduces novel file and node-level retrieval metrics based on syntax tree analysis. These metrics complement the standard execution-based "resolve rate", providing insights into the ability of an agent to navigate complex codebases. We carry out an extensive set of experiments on SWE-PolyBench with leading open-source agents. Our results, summarized in Figure 1, demonstrate varying performance across different programming languages, with notable difficulties in tasks requiring complex, multi-file modifications or extensive code changes.

## 2 RELATED WORK

Code generation has a long history extending back to program synthesis (Manna & Waldinger, 1971; Gulwani et al., 2017). Here we focus on the recent work using large language models (LLM) for code generation (Jiang et al., 2024) and their corresponding methods for evaluation. Prior benchmarks for LLM based code generation can be broadly categorised into retrieval-free and retrieval-augmented, based on whether retrieval of salient snippets for editing is required prior to code generation. Several retrieval-free benchmarks have been proposed like APPS (Hendrycks et al., 2021), HumanEval (Chen et al., 2021), MBPP (Austin et al., 2021; Athiwaratkun et al., 2023), and MCoNaLa (Wang et al., 2022). Library specific datasets like NumpyEval (Zan et al., 2022) have been proposed to study LLMs when fine-tuned on those libraries' APIs. These benchmarks, providing all necessary information in the prompt, fail to simulate real-world software engineering tasks where identifying the location for edits is as crucial as determining the edits themselves.

Benchmarks can further be classified as execution-free or execution-based, depending on how code correctness is assessed. Execution-based benchmarks, more common among the aforementioned ones, use predefined tests to verify generated code. Among these, SWE-bench (Jimenez et al., 2024) is the first dataset that mirrors real-world software engineering, in which a code generation system is provided with a codebase and a problem statement, and is tasked to edit that codebase to solve the problem. SWE-bench was collected from GitHub issues of popular Python repositories. Tests for correctness are the unit tests automatically extracted from the repositories themselves. SWE-bench-java (Zan et al., 2024) comprises instances in Java, SWE-bench Multimodal includes visual elements from front-end tasks (Yang et al., 2024b), and Multi-SWE-Bench (Zan et al., 2025), most similar to our work, contains 1632 samples from seven languages. While Multi-SWE-Bench provides samples from additional languages, its evaluation procedure remains superficial, focusing primarily on pass rates and thus limiting insights into both agent performance and failure modes.

SWE-Lancer (Miserendino et al., 2025) includes web development and "manager-style" problems, where the task is to select among different implementations. Extending the topic of the problems being considered, Baxbench (Vero et al., 2025) is security focused and shows that most LLMs generate insecure code. SWE-PolyBench falls in the same category of execution-based datasets.

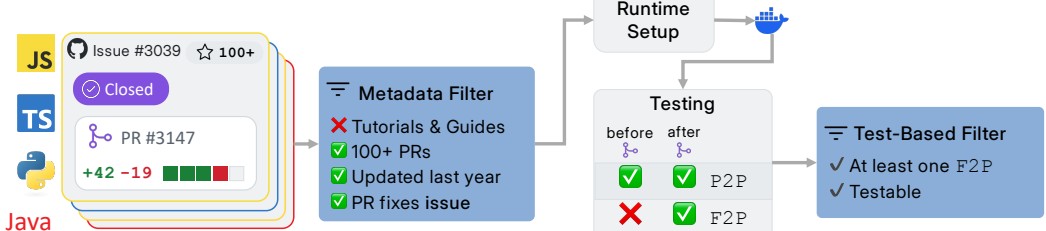

Figure 2: Overview of our dataset generation pipeline. We start by collecting pull requests (PRs) that close an issue from popular repositories across our four target languages. After applying a metadata filter, we then create containerized environments for test execution. We compare test outcomes before and after patch application. A test is fail-to-pass (F2P) if it initially fails but passes after applying the patch, other tests are pass-to-pass (P2P). The final test-based filter selects PRs that have at least one F2P test and are considered "testable" (condition detailed in Section 3).

It seeks to overcome some limitations of previous datasets, such as over-representation of single repositories, or lack of diversity in programming languages or type of tasks.

## 3 BUILDING SWE-POLYBENCH

In this section, we summarize the construction process of SWE-PolyBench, which involved first a metadata-based screening of public repositories, and then a runtime-based filtering to ensure feasibility of execution-based evaluation of proposed solutions through running (unit) test suites. Fig. 2 overviews our dataset collection pipeline. Tables 9 and 10 in the appendix provide an overview of the dataset statistics throughout the collection and filtering process.

Following the practice established by SWE-Bench, each instance of SWE-PolyBench includes a problem statement (which could be an issue or a PR description or combination of both), a reference to the repository, the "base commit" hash and the repository content, a "code patch" and a "test patch", which are git diffs of main source code and test suites, respectively. We postpone a broader formalization to Section 5.

**Data collection.** We built SWE-PolyBench using public GitHub repositories whose primary programming language is either Java, JavaScript (JS), Python, or TypeScript (TS) as these languages represent a significant portion of modern software development projects, according to Stack Overflow (2023). We set up the following rules as an initial filter: the repository A) is implementation-focused (namely we exclude guides, tutorials and similar repositories to focus on code); B) contains at least 100 pull requests (PRs), chosen as a threshold to indicate a non-trivial software project with an established history of collaborative development; C) was updated in the last 12 months to maintain relevance to current practices; D) is permissively licensed to allow unrestricted use for research; E) is in English (primarily), to facilitate analysis of code and discussions. Having collected a number of repositories that satisfy these rules, we consider a PR for inclusion into SWE-PolyBench if it solves an issue and provides respective test code that allows for execution-based verifiability. This process resulted in 17 repositories for Java, 12 for JS and 10 for Python and TS, yielding a total of 377 300 PRs. To ensure independent evaluation of model performance, we excluded any repositories found in SWE-Bench.

**Runtime setup.** We want to make sure that the correctness of proposed solutions are verifiable by executing a suite of unit tests. Thus, we filter PRs based on the following rules: 1) after applying the "test patch" to the base commit, the test suite includes at least one test that fails at base commit but passes after applying the "code patch"; following convention we call such set of tests "fail-to-pass"; 2) we exclude PRs that introduce and test new files in the code patch, as tests cannot reliably evaluate LLM-generated code when correct functionality appears in unexpected file locations or function names. Running these filters required creating Docker environments in which to execute tests, which we release alongside the evaluation harness. Further details in Appendix B.

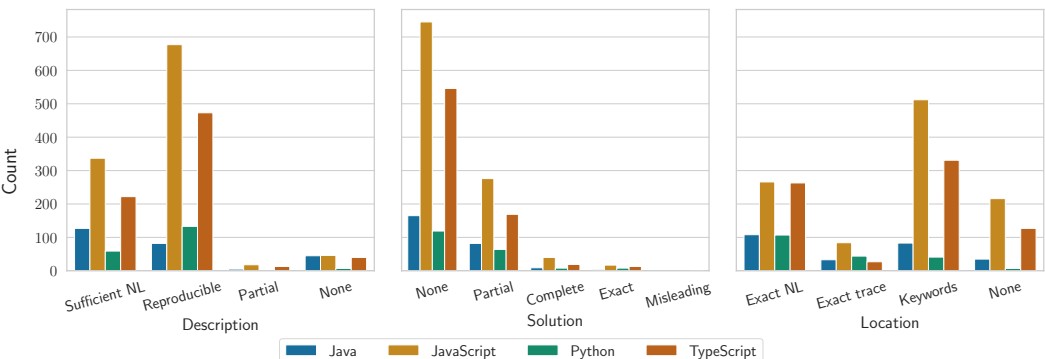

Figure 3: Classification of SWE-PolyBench issue descriptions with respect to their descriptiveness (left plot), hints at the solution (middle plot), and information on the localization of the issue (right plot). NL stands for natural language.

# 4 SWE-POLYBENCH CHARACTERISTICS

In this section, we cover the categorizations of SWE-PolyBench instances along three axes: the type of task, the informativeness level of the problem statement, and the task complexity. Next, we describe the curation of our verified version, SWE-PolyBench_Verified, partially informed by the aforementioned categorizations. Finally, we contrast our datasets with the established SWE-Bench and its "verified" subset, and discuss similarities and differences.

**Type of task.** Software issues encompass various tasks beyond bug fixes, such as refactoring requests or new feature implementations. We classified each problem statement into one of five categories: "bug fix", "feature request", "refactoring", "security" or "testing". We used as classifier a zero-shot prompted LLM, giving as context the problem statement, the code and test patches and and a task instruction which we report in Appendix E.2 and show the distribution of different task categories in Table 1.

**Informativeness of the problem statement.** We assess problem statements' "quality" along the dimensions and classes introduced by Xia et al. (2024). Specifically, we break down informativeness and quality alongside the following three dimensions: i) how descriptive is the problem statement, ii) how much information does the problem statement contain with respect to the desired solution, and iii) what localization information (e.g. file path) is available to address the described issue. For each of these dimensions, we consider four or five different classes to indicate informativeness levels, e.g. ranging from "full information" to "no information" (further details in Table 7). As above, we use a zero-shot prompted LLM to perform the three classification tasks, providing as context the problem statement and code and test patches and a prompt template we report in Appendix C.2. We provide descriptive statistics pertaining to the above classifications in Fig. 3. Importantly, we observe that only few instances provide the exact or complete solutions in the problem statement.

**Complexity of the task.** We measure task complexity in SWE-PolyBench through two primary metrics: (1) the number of files that need to be modified to implement a solution, and (2) the granularity and distribution of changes in the concrete syntax tree (CST) nodes, specifically focusing on modifications to class and function nodes. At the CST level, we categorize changes into four types: those requiring no class or function modifications (None), function-only changes, class-only changes, and mixed modifications affecting both classes and functions. Table 8 (middle rows) summarizes our analysis. Overall Java instances exhibit the highest complexity, requiring modifications to 3.6 files on average and showing mixed node changes in 66.06% of cases. Python tasks demonstrate moderate complexity, while JavaScript and TypeScript show distinct patterns - JavaScript has the highest proportion of function-only modifications, and TypeScript shows the highest percentage of non-class/function changes.

Table 1: Contrasting SWE-PolyBench (`PB`) and SWE-PolyBench_Verified (`PBv`) file statistics and task categories with SWE-Bench (`SWEb`) (Yang et al., 2024a) and SWE-Bench verified (`SWEv`) (Chowdhury et al., 2024).

| | Modified Files (avg.) | | | | Task Category (%) | | | |
|---|---|---|---|---|---|---|---|---|
| | Python | Java | JS | TS | All | Bug Fix | Feature Req. | Refac. | Misc. |
| `SWEb` | 1.6 | – | – | – | 1.6 | 75.50 | 18.40 | 5.14 | 0.96 |
| `SWEv` | 1.2 | – | – | – | 1.2 | 87.20 | 8.60 | 4.00 | 0.20 |
| `PB` | 2.0 | 3.6 | 2.2 | 3.1 | 2.6 | 74.50 | 21.94 | 2.94 | 0.61 |
| `PBv` | 1.5 | 3.1 | 2.0 | 3.1 | 2.4 | 59.80 | 14.0 | 2.6 | 0.00 |

Table 2: Dataset complexity in terms of changes in concrete syntax tree nodes. Highest numbers per column are highlighted in bold.

| Dataset | Language | Node Change Category (%) | | | | Node Change Count (avg.) | | |
|---|---|---|---|---|---|---|---|---|
| | | None | Func. only | Class only | Mixed | Func. | Class | Num. Nodes |
| `SWEb` | Py | 1.48 | 67.26 | 3.27 | 27.99 | 2.81 | 0.72 | 3.54 |
| `SWEv` | Py | 1.80 | 77.60 | 4.20 | 16.40 | 1.87 | 0.30 | 2.18 |
| `PB` | Py | 1.01 | 55.78 | **5.53** | 37.69 | 4.09 | 1.67 | 5.76 |
| | Java | 0.00 | 32.12 | 1.82 | **66.06** | 7.35 | **2.45** | **9.81** |
| | JS | 3.74 | **84.27** | 0.29 | 11.70 | 2.45 | 0.14 | 2.60 |
| | TS | **30.59** | 56.24 | 1.78 | 11.39 | 1.86 | 0.21 | 2.06 |
| | All | 12.46 | 67.82 | 1.42 | 18.29 | 2.78 | 0.49 | 3.28 |
| `PBv` | Py | 0.00 | 67.26 | 7.08 | 25.66 | 3.22 | 1.10 | 4.32 |
| | Java | 0.00 | 39.13 | 1.45 | 59.42 | **7.39** | 1.97 | 9.36 |
| | JS | 5.00 | 82.00 | 0.00 | 13.00 | 2.21 | 0.17 | 2.39 |
| | TS | 27.00 | 64.00 | 2.00 | 7.00 | 1.66 | 0.11 | 1.77 |
| | All | 8.38 | 65.18 | 2.88 | 23.56 | 3.30 | 0.75 | 4.06 |

## 4.1 SWE-POLYBENCH_VERIFIED

To create a high-quality subset for efficient experimentation, we enlisted expert annotators to evaluate the quality of issue descriptions, code, and test patches in our dataset. After removing entries that fell below our quality threshold and downsampling JavaScript and TypeScript instances to maintain a manageable dataset size, we curated a refined collection of 382 instances spanning four programming languages. This carefully selected subset enables more rapid and focused experimentation while preserving essential quality standards. We refer to this dataset as SWE-PolyBench_Verified (`PBv`). The details of the annotation process and our scoring criteria is described in Appendix A.

## 4.2 CONTRASTING SWE-POLYBENCH WITH SWE-BENCH

We compare key characteristics of SWE-PolyBench (`PB`) and SWE-PolyBench_Verified (`PBv`) with SWE-Bench (`SWEb`) and SWE-Bench verified (`SWEv`), as the latter two are widely recognized as the current standard for evaluating coding agent performance. Key comparison statistics alongside the axes discussed above are reported in Table 1 and Table 2. `PB` exhibits higher complexity in terms of modified files across all languages. Overall 63 % more files need to be edited to solve a task in `PB` compared to `SWEb`. The task categories are similarly distributed between `PB` and `SWEb`.

## 5 METRICS

We assess coding agents by calculating their pass rates and various retrieval metrics, which we formalize in this section. Let $\mathcal{X}$ be a space of finite sequences of strings. Given a problem statement $p \in \mathcal{X}$ and the contents of a repository $r \in \mathcal{X}$, an LLM-based agent $f_{\text{LLM}}$ outputs an updated

repository $r' = f_{\text{LLM}}(p, r) \in \mathcal{X}$ [1]. Furthermore, denote by $r' \setminus r \in \mathcal{X}$ the difference between the input repository and the edited repository.

**Pass rate.** We consider $p$ to be solved by the agent if the execution of a set of tests on $r'$ is successful. In line with previous work (Jimenez et al., 2024), the tests comprise a number of pass-to-pass (P2P) and fail-to-pass tests (P2P), as detailed in Section 3. Formally, this means that for each instance of SWE-PolyBench we provide two Boolean functions $t_{\text{P2P}} : \mathcal{X} \to \{0, 1\}$ and $t_{\text{F2P}} : \mathcal{X} \to \{0, 1\}$ such that $t_{\text{P2P}}(r) = 1$ and $t_{\text{F2P}}(r) = 0$. We let $t = t_{\text{P2P}} \wedge t_{\text{F2P}}$ be their conjunction. Then, we define the pass rate of an agent $f_{\text{LLM}}$ on a dataset $\mathcal{D}$ as follows:

$$\texttt{PassRate}(f_{\text{LLM}}, \mathcal{D}) = \frac{1}{|\mathcal{D}|} \sum_{(p,r,t) \in \mathcal{D}} t(f_{\text{LLM}}(p, r)). \tag{1}$$

**Retrieval scores based on code edits.** While pass rate is a central metric to measure code generation performance, it might fail to capture an agent's ability to successfully navigate a repository and localize relevant code elements. For this reason, we calculate file-level retrieval metrics (recall and precision) and introduce a new set of CST node-level retrieval metrics which we define below.

Let $r^* \in \mathcal{X}$ be the repository patched with the ground truth patch, and let $\texttt{F} : \mathcal{X} \to 2^{\mathcal{X}}$ be a set-valued function that given a diff string, extracts the file paths changed in the diff. Then, to compute file-level retrieval scores (e.g., recall and precision) we use $\texttt{F}(r^* \setminus r)$ as the ground truth set and $\texttt{F}(r' \setminus r)$ as the predicted set.

For node-level retrieval scores, we construct the CST for each changed file of a given `diff` string. A CST is a detailed tree representation of the source code that preserves syntactic details while representing structural elements like functions and classes as nodes in the tree. We then retrieve the modified nodes (i.e., `module`, `class`, `function`) accounting for their depth in the tree. More formally, let $\texttt{CST} : \mathcal{X} \times \mathcal{X} \to \mathcal{T}$ be the parsing function that given a `diff` string and a repository, returns a labeled tree representing the portion of the CST touched by the `diff`. The labels of each node uniquely identify a section of the repository (e.g. `class`, or `function`), but do not include their actual content. Let $\texttt{Paths} : \mathcal{T} \to 2^{\mathcal{T}}$ be a function that extract all root-to-leaf paths of a tree, yielding a set of paths (i.e., trees with degree at most 1). Then, to compute node-level retrieval scores we consider $\texttt{Paths}(\texttt{CST}(r^* \setminus r, r^*))$ to be the ground truth set and $\texttt{Paths}(\texttt{CST}(r' \setminus r, r'))$ to be the predicted set. For details about CST construction, please refer to Appendix D.

## 6 EVALUATING OPEN-SOURCE CODING AGENTS

We run a series of experiments with open-source agents on SWE-PolyBench and SWE-PolyBench_Verified to provide a snapshot of current performances on our newly introduce datasets. We start the section by describing the code agents we chose and then discuss experimental results, focusing on the metrics introduced above.

**Coding agents.** For our comparison, we selected three open-source agents that are widely recognized and appreciated in both the research community and among practitioners:

- **Aider** (Gauthier, 2024) (v0.75.2), an interactive pair programming agent. The agent suggests different changes to the codebase and the user can select or submit their preferences. For benchmarking, we run Aider in non-interactive mode.

- **SWE-agent** (Yang et al., 2024a) (v1.0) which employs an agent-computer interface that can create and edit code files, navigate entire repositories, and execute tests and other programs.

- **Agentless** (Xia et al., 2024) (v1.5.0) which uses a three-phase approach to 1) localize, 2) repair, and 3) validate code. Agentless does not rely on autonomous agent-based interactions with tools.

We modified and adapted these agents to address the specific challenges of SWE-PolyBench, resulting in their modified versions: `Aider-PB`, `SWE-Agent-PB`, and `Agentless-PB`. In summary:

---

[1] For simplicity, we take here the agent to be deterministic. In practice, agents are typically stochastic.

Table 3: Pass rates of open source agents on SWE-PolyBench.

| Agent | Base LLM | Language | | | | Overall |
|---|---|---|---|---|---|---|
| | | Java | JS | TS | Python | |
| Agentless-PB | Sonnet 3.5 | $10.9_{\pm2.44}$ | $7.2_{\pm0.81}$ | $4.7_{\pm0.77}$ | $20.1_{\pm2.95}$ | $7.8_{\pm0.59}$ |
| SWE-Agent-PB | Sonnet 3.5 | $\mathbf{16.4_{\pm2.88}}$ | $6.5_{\pm0.77}$ | $10.2_{\pm1.10}$ | $\mathbf{24.1_{\pm3.05}}$ | $10.2_{\pm0.66}$ |
| Aider-PB | Sonnet 3.5 | $15.8_{\pm2.86}$ | $\mathbf{12.6_{\pm1.04}}$ | $\mathbf{13.0_{\pm1.24}}$ | $24.1_{\pm3.05}$ | $\mathbf{14.1_{\pm0.77}}$ |
| | Deepseek R1 | $12.1_{\pm2.55}$ | $10.1_{\pm0.95}$ | $11.5_{\pm1.17}$ | $18.1_{\pm2.81}$ | $11.5_{\pm0.71}$ |
| | Haiku | $11.5_{\pm2.49}$ | $8.1_{\pm0.85}$ | $9.7_{\pm1.09}$ | $18.1_{\pm2.81}$ | $9.9_{\pm0.65}$ |
| | Mistral-Large | $6.7_{\pm1.96}$ | $4.8_{\pm0.66}$ | $6.9_{\pm0.93}$ | $7.0_{\pm1.83}$ | $5.9_{\pm0.51}$ |
| | Llama 3.3 70B | $9.1_{\pm2.24}$ | $4.2_{\pm0.62}$ | $6.4_{\pm0.91}$ | $11.1_{\pm2.27}$ | $6.0_{\pm0.52}$ |
| | DeepSeek-R1-Distill-Llama-70B | $5.5_{\pm1.78}$ | $3.5_{\pm0.57}$ | $5.8_{\pm0.86}$ | $12.6_{\pm2.40}$ | $5.3_{\pm0.48}$ |

1) we removed the validation step of **Aider (aider-swe-bench)**, as the original implementation is tailored to Python projects; 2) we modified **SWE-agent**'s containerized environment by creating custom Docker configurations and using Javascript base images to resolve package compatibility issues; 3) we adapted **Agentless** to support multiple languages by replacing Python-specific tools with tree-sitter and implementing language-specific execution commands. We refer the reader to Appendix E.1 for further details on technical challenges of adapting these agents to multi-language settings. All implementations utilize `Anthropic's Claude 3.5` (claude-3-sonnet-20241022) as the foundation large language model if not stated otherwise.

## 6.1 RESULTS

**Pass rates.** In Tables 3 to 5, we examine pass rates across programming languages, task complexity, task categories as well as token efficiency. Performance stratified by task types is reported in Appendix E.2. In addition, Table 6 summarizes both file and CST node retrieval accuracy to get a fine-grained view of the agents' capability to navigate the code repository. If not indicated otherwise, we report the mean pass rate with associated standard error ($\%_{\pm SE}$), estimated via bootstrap resampling over $n = 2000$ iterations. Fig. 1 and Tables 3 and 8 reveal significant performance variations across programming languages and change types for the three Sonnet 3.5-based agents. All agents demonstrate their strongest performance in Python (20 % to 24 %), however these rates remain relatively modest compared to pass rates in SWE-bench (Jimenez et al., 2024). Performance in Java (11 % to 16 %) and particularly TypeScript (5 % to 13 %) is strikingly lower than the other two languages. These findings suggest that pass rates stem from a complex interplay between task complexity, node change types, and language-specific factors that likely reflect the distribution of programming languages and structural patterns in LLMs' pretraining data. Stratifying problems by their complexity (Table 8), models perform best on "class only" and "single class" modifications (25 % to 40 %), while degrading significantly with "mixed" changes (8 % to 15 %). Surprisingly, "function only" and "single function" changes also yield relatively low success rates (around 15 %), despite their typically more contained scope. Table 4 breaks down performance by task category. `Aider-PB` achieves the highest average pass rates, while being more token efficient than other agents. Table 5 reveals how performance degrades with increasing task complexity, where all methods reach their maximal pass rate on single file edits.

On SWE-PolyBench_Verified, `Aider-PB` (sonnet 3.5) also achieves the highest average pass rate (16.23%), with `SWE-Agent-PB` scoring 14.4% and `Agentless-PB` scoring 13.35%.

**Retrieval metrics.** Table 6 presents an evaluation of file and node retrieval metrics, demonstrating varying efficacy of different agent-model combinations. In file retrieval, performance varies significantly across languages, with `SWE-Agent-PB` achieving the highest recall in Java (51.6 %), while `Aider-PB` with Sonnet 3.5 excels in precision (65.1 %). For both JavaScript and TypeScript, `Aider-PB` performs significantly better than other agents in both precision and recall. Notably, while `Agentless-PB` outperforms all other configurations in Python file and node retrieval, its strength is limited to Python instances alone. Node retrieval results follow similar patterns, with `Aider-PB` leading in Java, JavaScript, and TypeScript, and `Agentless-PB` maintaining supe-

Table 4: Average pass rates with standard error by task category and average token usage per instance.

| Agent | Base LLM | Category | | | Tokens (avg.) | |
|---|---|---|---|---|---|---|
| | | Bug Fix | Feature Req. | Refac. | Input | Output |
| Agentless-PB | Sonnet 3.5 | $8.8_{\pm 0.71}$ | $5.2_{\pm 1.03}$ | $3.2_{\pm 2.27}$ | 315 461 | 10 900 |
| SWE-Agent-PB | Sonnet 3.5 | $10.2_{\pm 0.76}$ | $9.5_{\pm 1.34}$ | $\mathbf{16.1_{\pm 4.70}}$ | 338 828 | 679 |
| Aider-PB | Sonnet 3.5 | $\mathbf{13.8_{\pm 0.89}}$ | $\mathbf{15.1_{\pm 1.66}}$ | $12.9_{\pm 4.25}$ | 64 521 | 845 |
| | Deepseek R1 | $11.7_{\pm 0.82}$ | $10.4_{\pm 1.40}$ | $\mathbf{16.1_{\pm 4.70}}$ | 53 440 | 2367 |
| | Haiku | $9.9_{\pm 0.75}$ | $9.9_{\pm 1.37}$ | $9.7_{\pm 3.72}$ | 64 067 | 1094 |
| | Mistral-Large | $5.7_{\pm 0.57}$ | $6.5_{\pm 1.14}$ | $4.8_{\pm 2.77}$ | 63 946 | 7319 |
| | Llama 3.3 70B | $6.6_{\pm 0.62}$ | $4.1_{\pm 0.93}$ | $4.8_{\pm 2.77}$ | 84 311 | 2002 |
| | DeepSeek-R1-Distill-Llama-70B | $5.9_{\pm 0.58}$ | $3.2_{\pm 0.81}$ | $6.5_{\pm 3.14}$ | 58 028 | 2241 |

Table 5: Performance of different open-source agents on SWE-PolyBench with varying task complexity in terms of files edited. Number of instances in the dataset are in parenthesis.

| Agent | Base LLM | Files to be modified | | | | |
|---|---|---|---|---|---|---|
| | | 1 (1085) | 2 (417) | 3 (200) | 4 (130) | 5+ (278) |
| Agentless-PB | Sonnet 3.5 | $10.8_{\pm 0.94}$ | $7.0_{\pm 1.24}$ | $4.0_{\pm 1.38}$ | $2.3_{\pm 1.33}$ | $2.9_{\pm 1.02}$ |
| SWE-Agent-PB | Sonnet 3.5 | $12.5_{\pm 1.01}$ | $10.6_{\pm 1.49}$ | $5.0_{\pm 1.56}$ | $3.8_{\pm 1.71}$ | $7.2_{\pm 1.56}$ |
| Aider-PB | Sonnet 3.5 | $\mathbf{17.7_{\pm 1.18}}$ | $\mathbf{13.9_{\pm 1.72}}$ | $\mathbf{8.0_{\pm 1.89}}$ | $6.9_{\pm 2.27}$ | $\mathbf{7.9_{\pm 1.63}}$ |
| | Deepseek R1 | $14.9_{\pm 1.09}$ | $9.8_{\pm 1.45}$ | $6.0_{\pm 1.68}$ | $\mathbf{7.7_{\pm 2.35}}$ | $6.5_{\pm 1.51}$ |
| | Haiku | $12.9_{\pm 1.02}$ | $8.6_{\pm 1.37}$ | $4.0_{\pm 1.38}$ | $6.2_{\pm 2.17}$ | $5.8_{\pm 1.43}$ |
| | Mistral-Large | $7.5_{\pm 0.79}$ | $5.0_{\pm 1.08}$ | $4.5_{\pm 1.47}$ | $2.3_{\pm 1.33}$ | $3.6_{\pm 1.13}$ |
| | Llama 3.3 70B | $8.3_{\pm 0.84}$ | $6.5_{\pm 1.21}$ | $2.5_{\pm 1.11}$ | $0.8_{\pm 0.78}$ | $1.4_{\pm 0.72}$ |
| | DeepSeek-R1-Distill-Llama-70B | $7.5_{\pm 0.79}$ | $4.6_{\pm 1.02}$ | $2.0_{\pm 0.99}$ | $1.5_{\pm 1.07}$ | $2.2_{\pm 0.88}$ |

riority in Python tasks. In general, we observe a significant gap between Python and the other languages. The highest file retrieval metrics in Python are ahead of the highest metric for any other language by 9.3 p.p. (percentage point) and 12.5 p.p. for recall and precision, respectively. The same holds for node retrieval, where Python metrics are ahead by 7.7 p.p. and 12.5 p.p. for recall and precision, respectively. Lastly, we would like to stress that, as evidenced by the pass rate of Agentless-PB, high retrieval metrics are (most of the time) a necessary but not sufficient condition for high pass rates.

## 7 CONCLUSIONS AND LIMITATIONS

We introduced SWE-PolyBench, a repository-level, multi-language benchmark for execution-based evaluation of coding agents. SWE-PolyBench comprises 2110 samples from 21 repositories across Java, JavaScript, TypeScript, and Python, covering bug fixes, feature requests, and code refactoring. We also provided SWE-PolyBench_Verified, an annotated high-quality subset for efficient experimentation. Our evaluation of leading open-source coding agents required significant effort to adapt the agents to multiple languages and revealed significant variations of performance across languages in terms of both pass rate and navigation proficiency (observed through our introduced retrieval metrics). Together, these factors stress the (over)-specialization to the Python ecosystem of several current solutions. Our categorization of the datasets along CST-rooted complexity axes revealed a consistent decline in performance as task complexity increased. Our findings underscore the need for more versatile and robust AI coding assistants capable of handling complex real-world software engineering tasks across multiple programming languages. SWE-PolyBench aims to drive progress in developing such agents by providing a comprehensive, multi-lingual evaluation framework.

Table 6: File and node retrieval metrics for different open-source agents on SWE-PolyBench.

| Agent | Base LLM | File Retrieval (%) | | | | | | | |
| | | Java | | JavaScript | | TypeScript | | Python | |
| | | Recall | Precision | Recall | Precision | Recall | Precision | Recall | Precision |
|---|---|---|---|---|---|---|---|---|---|
| `Agentless-PB` | Sonnet 3.5 | 29.5 | 49.7 | 23.4 | 35.2 | 17.5 | 27.7 | **60.9** | **77.6** |
| `SWE-Agent-PB` | Sonnet 3.5 | **51.6** | 58.5 | 27.5 | 28.5 | 29.8 | 36.4 | 59.7 | 44.2 |
| `Aider-PB` | Sonnet 3.5 | 41.7 | **65.1** | **37.1** | **53.5** | 33.3 | **52.0** | 58.2 | 73.8 |
| | Deepseek R1 | 37.6 | 53.8 | 31.5 | 40.8 | **33.8** | 46.0 | 54.7 | 63.3 |
| | Haiku | 35.0 | 53.2 | 28.3 | 40.3 | 30.2 | 45.1 | 56.8 | 70.3 |
| | Mistral Large | 30.6 | 46.8 | 21.6 | 30.9 | 25.2 | 38.5 | 47.6 | 55.9 |
| | Llama 3.3 70B | 27.7 | 43.3 | 20.6 | 28.9 | 24.7 | 39.6 | 42.9 | 54.5 |
| | DeepSeek-R1-Distill-Llama-70B | 31.9 | 47.0 | 25.0 | 31.7 | 27.1 | 36.2 | 48.7 | 59.9 |
| | | Node Retrieval (%) | | | | | | | |
| `Agentless-PB` | Sonnet 3.5 | 20.6 | 38.9 | 18.9 | 27.6 | 17.2 | 22.9 | 38.2 | **63.6** |
| `SWE-Agent-PB` | Sonnet 3.5 | **32.5** | **52.3** | 28.8 | 23.7 | 21.7 | 20.6 | **38.6** | 61.1 |
| `Aider-PB` | Sonnet 3.5 | 29.2 | 51.1 | **30.5** | **39.9** | 20.2 | 26.7 | 36.7 | 59.9 |
| | Deepseek R1 | 24.6 | 40.2 | 26.2 | 31.2 | **23.6** | **29.4** | 33.5 | 50.9 |
| | Haiku | 22.8 | 38.4 | 22.8 | 29.1 | 18.2 | 23.5 | 33.0 | 53.8 |
| | Mistral Large | 24.3 | 21.3 | 17.5 | 16.4 | 15.3 | 15.9 | 38.1 | 16.7 |
| | Llama 3.3 70B | 18.7 | 31.5 | 15.0 | 19.4 | 15.0 | 18.9 | 24.4 | 39.5 |
| | DeepSeek-R1-Distill-Llama-70B | 20.5 | 32.5 | 19.4 | 22.0 | 17.8 | 20.5 | 28.1 | 44.9 |

**Limitations and societal impact.** We conclude the work with a discussion of limitations, societal impact of SWE-PolyBench and potential future directions.

*Task diversity*: there is a "long tail" of problems that are part of the day-to-day work of software developers that are not addressed in this benchmark.We believe that targeting the "head of the distribution" of tasks is a good first step, but future work should consider expanding to cover a broader range of software engineering challenges.

*Evaluation Metrics*: our evaluation metrics do not capture several aspects of code quality and correctness, such as adherence to code best practices or repository style guides, maintainability, or the presence of potential security flaws in the generated code. Providing a more holistic assessment of coding agent performance – remains a challenging direction for future work.

*Limits of execution-based evaluation*: execution-based evaluation using (unit) test suites is the de-facto standard for coding benchmarks, providing a quick and cheap feedback signal. However, it may also constrain the type of tasks one may be able to reasonably verify and hardly accounts for completely valid variations (e.g., in class, function, and variable naming). Going beyond the current practice remains an open area of research.

*Verifiability*: another important limitation of SWE-PolyBench is the lack of human verification to ensure that all tasks are "solvable" based on the provided information. Future benchmarks should strive to balance the need for verifiable tasks with the preservation of diverse issue description qualities, mirroring the range of scenarios encountered in practical software development.

*LLM-based classifications*: for our analysis of task type and informativeness of the task description, we made extensive use of LLM-based zero-shot classifiers. Our annotations provide complementary information that can guide the development of specialized approaches and adds another dimension to the evaluation which informs about existing gaps. However, LLM-based annotations are not without risk as pointed out in Ahmed et al. (2024) and should be interpreted accordingly.

*Data leakage*: the publicly available data used to create SWE-PolyBench may have been utilized during training of foundational LLMs we used in evaluation, or might be used in the future. Data leakage concerns and lack of transparency creates an ever-shrinking window to develop truly novel evaluation data, underscoring the need for innovative approaches to evaluation and benchmarking, e.g. test-set slot guessing (Deng et al., 2023).

## 8 REPRODUCIBILITY STATEMENT

We have uploaded our evaluation source code along with the dataset files under `datasets/` inside the zip file in the supplement materials. The instructions to run the evaluation and get the pass rates with retrieval metrics can be found in `README.md`. The steps to install the dependencies can be found in `requirements.txt`.

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

## A    CURATING SWE-POLYBENCH_VERIFIED

**Annotation.**    We enlisted expert annotators to score the instances from our datasets on different metrics. Our annotators are software developers with experience in the programming language they were tasked to annotate. They are asked the following questions per instance:

1. **Issue Quality:** I have sufficient information to implement a solution.
   - Scores range from 0 to 3, with 0 being the lowest quality score

2. **Code Quality:** Which of the functionalities from the issue description are covered in the code changes?
   - Scores range from 0 to 3, with 0 being changes unrelated to the issue

3. **Test Quality:** How effectively do the tests evaluate solutions for the described issue?
   - Scores range from 0 to 3, with 0 and 1 being prone to false positive/false negatives, respectively. Score 2 means generally reliable test, 3 indicates highly accurate tests

**Filtering.**    We applied the following filtering criteria to ensure high-quality instances:

- **Issue Quality:** All raters have assigned a score of at least 2 (i.e., "I have most of the required information and can infer any minor missing details to proceed with implementation" (2) or "I have all the necessary details and context to create a meaningful solution without any additional information" (3))

- **Code Quality:** All raters have assigned a score of exactly 2 (i.e., "All specified functionality is implemented exactly as described")

- **Test Quality:** All raters have assigned a score of at least 2 (i.e., "Generally Reliable: Tests are effective in most cases but may occasionally produce false positives (passing incorrect solutions) or false negatives (failing correct solutions), particularly with edge cases or less common scenarios" (2) or "Highly Accurate: Tests consistently identify correct solutions and reject incorrect ones" (3))

This filtering process resulted in 382 instances (69 in Java, 113 in Python, 100 in JS, 100 in TS) from 20 repositories.

## B    RUNTIME SETUP.

In addition to the filtering mentioned in  Section 3, we assess whether a PR can be included in the dataset through execution. In the following we refer to "test patch" and "code patch" which are defined as follows: The code patch is the `git diff` containing changes in the PR that do not relate to the tests. We refer to this patch also as the "ground truth patch" as it is the ground truth code implementation which solves the issue. The test patch is the `git diff` containing only changes relating to test updates. Additionally, the "base commit" is the commit onto which the respective PR was merged.

For each PR, to which we also refer to as task, a Docker file is defined to serve as the execution context in which the code base is installed prior to applying any patches. Given that each programming language has its own package manager, installation procedures, and version requirements, we tailored the setup accordingly. For example, Java projects commonly use `Maven` for project management and build automation, while JavaScript and TypeScript projects typically rely on `npm` for package and dependency management. For each repository and/or base commit, we manually configured Docker files to set up the execution environment. Within this environment, we ran the test suite both before and after applying the code patch. We then logged two sets of tests: those that transitioned from a 'failed' state to a 'passed' state (which we refer to as `F2P` or fail-to-pass), and those that passed both before and after the application of the code patch (which we call `P2P` or pass-to-pass).

For a PR to be included in the dataset it must contain at least one `F2P` test. Lastly, we deem PRs untestable and exclude them if the code patch introduces new files whose contents are tested

in the code patch. This is because created tests cannot reliably evaluate LLM-generated code if functionally correct code was created in unexpected file or function names. To run execution-based evaluation, all Docker files are published alongside our evaluation harness.

Tables 9 and 10 in the appendix provide an overview of the dataset statistics throughout the collection and filtering process. The average repository size varies significantly across languages, with TypeScript repositories being the largest on average (8946.0 files) and Python repositories the smallest (1928.1 files). Note that these counts include binary and documentation files.

## C PROMPTS

### C.1 PROMPT FOR CLASSIFICATION OF TASKS

```
prompt = """
   You will be provided with a problem description provided by a
      ↪ user to a github repository, which is labeled as an issue
      ↪  in github, along with the patch that solves the problem.
      ↪  Your task is to try to classify the problem in to a
      ↪ category from the list of categories provided below.
   Problem description:
   {{problem_statement}}
   The gold patch is a diff file that addresses the changes made
      ↪ to the files in the repository in order to solve the
      ↪ issue. It contains the list of files modified or added or
      ↪  removed and the code lines that have been added or
      ↪ replaced or removed.
   Following is the gold patch in a diff format that solves the
      ↪ issue:
   {{gold_patch}}

   This is the list of classes we want to classify into, each with
      ↪  description of which issue would the class as its label:
   "Bug Fix": "the problem asks for addressing bugs or issues
      ↪ reported",
   "Feature": "the problem is about introducing new features or
      ↪ enhancements",
   "Testing": "the problem is about adding new testing methods for
      ↪  given code or refactoring existing tests. These could be
      ↪  unit or integration (e2e) tests",
   "Refactoring": "the problem suggests to refactor existing code
      ↪ without changing its external behavior. This could
      ↪ include improving code readability, performance
      ↪ optimizations, or restructuring code for better
      ↪ maintainability",
   "Security": "the problem asks to address security
      ↪ vulnerabilities or concerns in the codebase",

   You should output the selected class in the XML format
      ↪ mentioned below. You should only classify into exactly
      ↪ one class. An example output will look like:
   ```
   <category>Feature</category>
   ```
   You must not include any additional text other than the XML and
      ↪  no additional XML tags as well. The value within XML
      ↪ tags should be exactly the same as one of the categories:
      ↪  Bug Fix, Feature, Refactoring.
   """
```

## C.2  PROMPTS FOR CLASSIFICATION OF PROBLEM STATEMENTS

```
prompt = """
  Your job is to do the following three things:
  1. You will classify the issues description according to its
      ↪ level of detail.
  2. You will classify the issues description according to
      ↪ whether it is solvable given the provided information.
  3. You will classify the issues description according to
      ↪ whether it mentions precise code locations to be changed.

  For tasks 1 to 3 I will also provide you with the correct
      ↪ solution of the problem, termed ground truth.

  Here is a detailed description of the tasks:

  ### TASK 1 ###
  You will assess if a github issue description is sufficiently
      ↪ detailed such that a software engineer can implemented
      ↪ the solution for the issue after inspecting the code base
      ↪ .

  You will have access to the issue description as well as a
      ↪ ground truth code patch, that is the desired solution.
      ↪ You will not have access to the code base itself.

  You will provide a brief explanation for your decision and then
      ↪  provide a label, 'A', 'B', 'C' or 'D' in the XML tags.
  Here are the different lables that you will use:
  'A' contains enough information in natural language to solve
      ↪ the issue
  'B' contains a reproducible failure example
  'C' contains a partially reproducible example
  'D' does not contain enough information to solve the issue

  Here is the format of the output:

  <explanation_description>YOUR_EXPLANATION</
      ↪ explanation_description>
  <label_description>YOUR_LABEL</label_description>

  ### TASK 2 ###

  You will help me evaluating the quality of a github issues
      ↪ description together with a ground truth patch that
      ↪ solves the described problem.
  In particular, your tasks is to check whether the solution or
      ↪ steps to solve the problem are already provided in the
      ↪ issue description.

  You will have access to the issue description as well as a
      ↪ ground truth code patch, that is the desired solution.
      ↪ You will not have access to the code base itself.

  You will provide a brief explanation for your decision and then
      ↪  provide a label, 'A', 'B', 'C', 'D' or 'E' in the XML
      ↪ tags.
  Here are the different lables that you will use:
  'A' no solution or steps provided
```

```
‘B‘ partial solution provided (e.g., some steps in natural
    ↪ language)
‘C‘ complete solution provided (e.g., complete steps in natural
    ↪  language)
‘D‘ exact patch provided
‘E‘ misleading solution or steps

Here is the format of the output:

<explanation_solution>YOUR_EXPLANATION</explanation_solution>
<label_solution>YOUR_LABEL</label_solution>

### TASK 3 ###

You will help me evaluating the quality of a github issues
    ↪ description together with a ground truth patch that
    ↪ solves the described problem.
In particular, your tasks is to check whether the issue
    ↪ description contains information on the issue location, i
    ↪ .e., which part of the code
needs to modified or fixed to address the issue.

You will have access to the issue description as well as a
    ↪ ground truth code patch, that is the desired solution.
You will not have access to the code base itself.

You will provide a brief explanation for your decision and then
    ↪  provide a label, ‘A‘, ‘B‘, ‘C‘, ‘D‘
in XML tags. You have to assign exactly one label per issue
    ↪ description.
Here are the different lables that you will use:
‘A‘ exact locations in natural language provided
‘B‘ exact locations provided in failure stack traces
‘C‘ related keywords in the issue description are provided that
    ↪  can be used to search for the location
‘D‘ no location provided.

Here is the format of the output:

<explanation_location>YOUR_EXPLANATION</explanation_location>
<label_location>YOUR_LABEL</label_location>
"""
```

## D  DETAILS CST RETRIEVAL METRICS

For node-level retrieval, we identify the deepest node of the concrete syntax tree (CST) accompanying a change. Let a CST be defined by the tuple $(\mathcal{V}, \mathcal{E}, \mathcal{L}, r, \lambda, \sigma)$, where $\mathcal{V}$ is the set of all vertices, $\mathcal{E} \subseteq \mathcal{V} \times \mathcal{V}$ the set of directed edges, $\mathcal{L}$ a finite set of node labels (e.g., class, function), $r \in \mathcal{V}$ the root node (e.g., a python module), and $\lambda$ a map $\lambda : \mathcal{V} \to \mathcal{L}$ assigning a label to a node. Furthermore, let a line span be defined as an interval $S := [s, e]$ where $s, e \in \mathbb{N}^+$ and $s < e$. Lastly, $\sigma : \mathcal{V} \to S$ is a map assigning a line span to a node. Since the line span of a higher-level node encompasses the spans of their descendants (e.g., a function's span lies in the interval of its parent's class's span), we want to identify the deepest node that is affected by a change. Formally, let $c \in S$ be the line span of a change on the code base. All nodes affected by a given change are defined as the set of nodes overlapping with the change (they have a non-empty overlap):

$$\text{affected}(c, \text{CST}) := \{v \in \mathcal{V} | c \cap \sigma(v) \neq \emptyset\}.$$

The deepest node affected by change $c$ can then defined as

$$\text{deepestNode}(c, \text{CST}) := v \in \text{affected}(c, \text{CST}) | \nexists u \in \text{affected}(c, \text{CST}) : \sigma(u) \subset \sigma(v).$$

In other words, the deepest node is the one among all affected nodes that does not contain another affected node.

To compute node-level retrieval metrics, we obtain $\mathcal{N}^i$ as the set of deepest nodes modified in the ground truth patch of task $i$, and $\hat{\mathcal{N}}^i$ as the set of deepest nodes affected by the predicted code changes for the task. Note, that for simplicity, we omitted indexing CST and changes with a file name. Naturally, we assume that the CST was constructed for the file in which changes were made.

# E   FURTHER EXPERIMENTAL DETAILS

## E.1   TECHNICAL CHALLENGES IN MAKING CODING AGENTS MULTI-LINGUAL

**Aider**   During its execution pipeline, Aider (v0.75.2) includes a validation step that runs preexisting tests against the model-generated patch to ensure it doesn't introduce regressions. This step requires two key components: access to the test execution command and a parser to interpret test results. In the original implementation, these components are specifically tailored for Python projects, utilizing `pytest` as the testing framework. However, adapting this process for SWE-PolyBench presents significant challenges. First, it would require maintaining a comprehensive database of test execution commands for each instance. Second, we would need to develop robust log parsers capable of interpreting test results across diverse testing frameworks. Given these complexities, we opted to exclude this validation step in `Aider-PB`.

**Agentless**   The original Agentless (v1.5.0) implementation employs Python-specific tools for its fault localization process, primarily using the `ast` python module to identify files, functions, and classes requiring modifications, as well as for linting. This Python-centric approach, however, limits its applicability to other programming languages. Similarly, its bug reproduction mechanism relies on generating and executing Python scripts, which is not generalizable across different languages. Furthermore, Agentless encounters the same regression testing limitations as Aider. In our adaptation, `Agentless-PB`, we address these limitations by incorporating `tree-sitter` for parsing and extracting code structures across JavaScript, TypeScript, and Java. We also implement language-specific execution commands for bug reproduction scripts. As with `Aider-PB`, we exclude the regression testing step from the pipeline.

**SWE-agent**   The original SWE-agent (v1.0) implementation relies on a containerized environment using `SWE-ReX` for interacting with repository contents. While SWE-agent supports custom Docker images, our adaptation process revealed significant compatibility challenges. It imposes specific requirements on pre-installed packages, including `Python3.11`, `SWE-ReX`, and `pipx`, within the provided Docker images. In our adaptation, `SWE-Agent-PB`, we initially addressed these issues by directly installing the missing packages in our Docker images, which resolved problems for a subset of instances. For the remaining cases, we explored an alternative approach: building a new Docker image on top of our provided image with a standalone Python installation. This method successfully isolates the required packages from the base image for many instances. However, it fails for 129 instances due to version incompatibilities between system libraries. For example, the new Docker image requires a specific `glibc` library version while some of our images contained older versions. For these 129 instances, we used the Javascript base image as generic image, which has a comprehensive list of pre-installed packages and also meets SWE-agent's package requirements. We then provided it as a custom docker image to run SWE-agent. Among the 129 instances, we obtained predictions for 111 instances. We treated the remaining 18 instances as empty predictions in `SWE-Agent-PB` when reporting performance metrics.

## E.2   TASK CLASSIFICATIONS AND PASS RATES

Following the categories in Xia et al. (2024) we use the categories in Table 7 for our classification, roughly ordered with respect to their level of information content. Fig. 4 shows the distribution of task categories on the sub-sampled SWE-PolyBench500 dataset.

Table 7: Informativeness of the problem statements

| Category | Description of the categories |
|---|---|
| *Descriptiveness of the problem statement* | |
| A | Contains enough information in natural language to solve the issue |
| B | Contains a reproducible failure example |
| C | Contains a partially reproducible example |
| D | Does not contain enough information to solve the issue |
| *Solution content already present in the problem statement* | |
| A | No solution or steps provided |
| B | Partial solution provided (e.g., some steps in natural language) |
| C | Complete solution provided (e.g., complete steps in natural language) |
| D | Exact patch provided |
| E | Misleading solution or steps provided |
| *Location information on the required changes* | |
| A | Exact locations in natural language provided |
| B | Exact locations provided in failure stack traces |
| C | Related keywords provided that can be used to search for the location |
| D | No location provided |

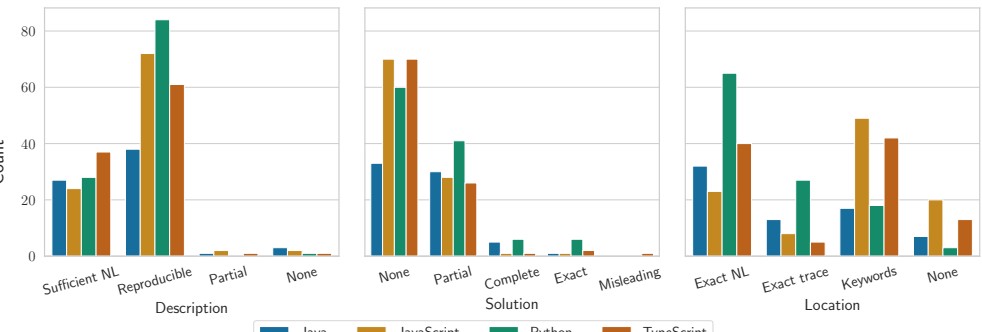

Figure 4: Classification of SWE-PolyBench_Verified issue descriptions with respect to their descriptiveness (left plot), hints a the solution (middle plot) and information on the localization of the issue (right plot

In Fig. 5 we highlight how the level of informativeness of the problem statements impact the pass rates across agents. Overall, more informative problem statements, be it with respect to location, hints at the solution or level of descriptiveness, result in higher pass rates. This confirms the intuition that less details in the problem statement make it more difficult for a task being solved.

### E.3 RESULTS ON SWE-POLYBENCH_VERIFIED

Fig. 6 presents the coding agents' performance across different programming languages and code change complexities. The left radar chart shows the pass rates for Java, JavaScript, TypeScript, Python, and overall performance. The right chart illustrates the agents' effectiveness in handling various types of code modifications, ranging from changes confined to a single class or function to more complex scenarios involving multiple structural elements.

## F COLLECTED REPOSITORIES

Tables 9 and 10 shows the repositories we collected and provides some statistics on the data collected.

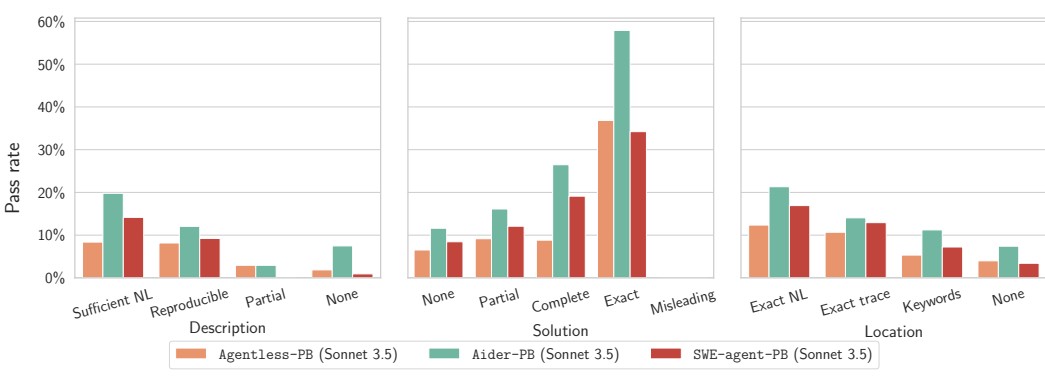

(a) Pass rates by task classification on SWE-PolyBench.

(b) Pass rates by task classification on SWE-PolyBench_Verified.

Figure 5: Pass rates of instances with respect to their descriptiveness (left plot), hints a the solution (middle plot) and information on the localization of the issue (right plot).

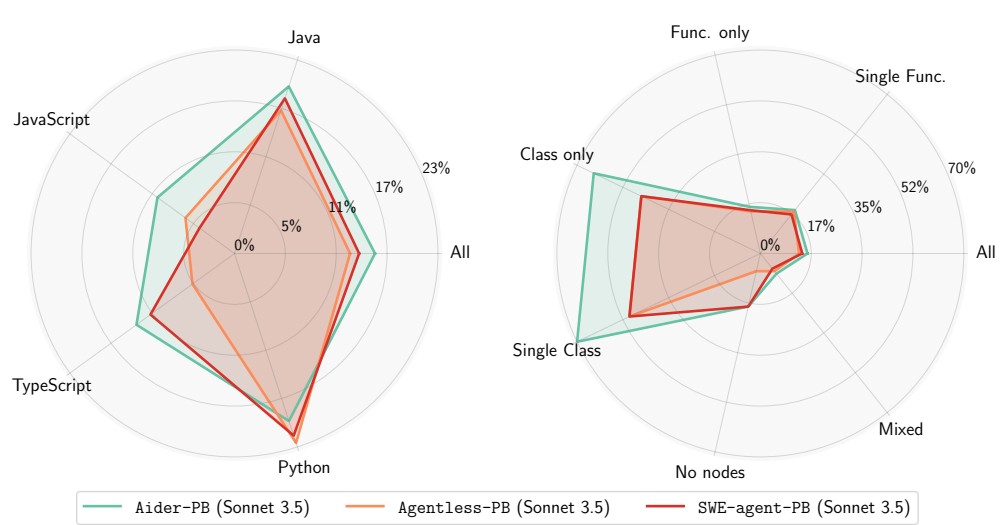

Figure 6: SWE-PolyBench_Verified pass rates of coding agents across programming languages (left) and across subsets of different complexities based on syntax tree nodes. The right plot categorizes changes by type (class or function) and scope (single or multiple), with "No nodes" indicating no class or function changes and "Mixed" requiring both.

Table 8: Pass rates of open source agents on SWE-PolyBench by complexity in terms of CST node changes.

| Agent | Base LLM | Node Change Category | | | | | |
| | | None $(n = 263)$ | Single Func. $(n = 848)$ | Func. Only $(n = 1431)$ | Single Class $(25)$ | Class Only $(n = 30)$ | Mixed $(n = 386)$ |
|---|---|---|---|---|---|---|---|
| Agentless-PB | Sonnet 3.5 | $3.8_{\pm 1.20}$ | $11.2_{\pm 1.08}$ | $8.7_{\pm 0.74}$ | $32.0_{\pm 9.42}$ | $26.7_{\pm 8.10}$ | $5.7_{\pm 1.18}$ |
| SWE-Agent-PB | Sonnet 3.5 | $14.4_{\pm 2.23}$ | $11.3_{\pm 1.09}$ | $9.6_{\pm 0.79}$ | $\mathbf{52.0_{\pm 9.96}}$ | $\mathbf{46.7_{\pm 9.12}}$ | $6.5_{\pm 1.26}$ |
| Aider-PB | Sonnet 3.5 | $\mathbf{19.8_{\pm 2.48}}$ | $\mathbf{17.0_{\pm 1.29}}$ | $\mathbf{13.8_{\pm 0.94}}$ | $40.0_{\pm 9.78}$ | $36.7_{\pm 8.93}$ | $\mathbf{9.3_{\pm 1.48}}$ |
| | Deepseek R1 | $17.5_{\pm 2.36}$ | $13.7_{\pm 1.18}$ | $10.8_{\pm 0.83}$ | $40.0_{\pm 9.78}$ | $36.7_{\pm 8.93}$ | $8.3_{\pm 1.41}$ |
| | Haiku | $16.3_{\pm 2.30}$ | $12.3_{\pm 1.12}$ | $9.3_{\pm 0.77}$ | $24.0_{\pm 8.58}$ | $23.3_{\pm 7.80}$ | $6.5_{\pm 1.26}$ |
| | Mistral-Large | $11.8_{\pm 2.02}$ | $7.1_{\pm 0.88}$ | $5.4_{\pm 0.58}$ | $20.0_{\pm 8.03}$ | $20.0_{\pm 7.41}$ | $2.6_{\pm 0.82}$ |
| | Llama 3.3 70B | $11.4_{\pm 1.98}$ | $7.4_{\pm 0.90}$ | $5.2_{\pm 0.58}$ | $32.0_{\pm 9.42}$ | $26.7_{\pm 8.10}$ | $3.6_{\pm 0.96}$ |
| | DeepSeek-R1-Distill-Llama-70B | $7.6_{\pm 1.63}$ | $7.1_{\pm 0.88}$ | $5.0_{\pm 0.57}$ | $28.0_{\pm 9.00}$ | $26.7_{\pm 8.10}$ | $3.1_{\pm 0.89}$ |

| Language | Repository | #PRs collected | License |
|---|---|---|---|
| Java | spring-projects/spring-boot | 6286 | Apache 2.0 |
| | PhilJay/MPAndroidChart | 378 | Apache 2.0 |
| | spring-projects/spring-framework | 4792 | Apache 2.0 |
| | google/guava | 2268 | Apache 2.0 |
| | NationalSecurityAgency/ghidra | 1044 | Apache 2.0 |
| | ReactiveX/RxJava | 3906 | Apache 2.0 |
| | apache/dubbo | 7165 | Apache 2.0 |
| | skylot/jadx | 536 | Apache 2.0 |
| | apolloconfig/apollo | 1676 | Apache 2.0 |
| | netty/netty | 7524 | Apache 2.0 |
| | Netflix/Hystrix | 760 | Apache 2.0 |
| | google/gson | 933 | Apache 2.0 |
| | libgdx/libgdx | 3576 | Apache 2.0 |
| | apache/rocketmq | 3794 | Apache 2.0 |
| | thingsboard/thingsboard | 5378 | Apache 2.0 |
| | JetBrains/intellij-community | 2518 | Apache 2.0 |
| | trinodb/trino | 16840 | Apache 2.0 |
| JavaScript | vercel/next.js | 22087 | MIT |
| | nodejs/node | 33429 | MIT |
| | axios/axios | 1482 | MIT |
| | mrdoob/three.js | 16172 | MIT |
| | facebook/react | 15393 | MIT |
| | twbs/bootstrap | 15096 | MIT |
| | sveltejs/svelte | 5324 | MIT |
| | atom/atom | 5249 | MIT |
| | angular/angular.js | 7928 | MIT |
| | lodash/lodash | 1383 | MIT |
| | prettier/prettier | 9613 | MIT |
| | serverless/serverless | 5557 | MIT |
| TypeScript | freeCodeCamp/freeCodeCamp | 36730 | BSD 3 clause |
| | microsoft/vscode | 30660 | MIT |
| | angular/angular | 27565 | MIT |
| | mui/material-ui | 22533 | MIT |
| | puppeteer/puppeteer | 5831 | Apache 2.0 |
| | storybookjs/storybook | 12461 | MIT |
| | tailwindlabs/tailwindcss | 2655 | MIT |
| | gothinkster/realworld | 795 | MIT |
| | supabase/supabase | 10743 | Apache 2.0 |
| | coder/code-server | 1863 | MIT |
| Python | Significant-Gravitas/AutoGPT | 3939 | MIT |
| | huggingface/transformers | 16135 | Apache 2.0 |
| | langchain-ai/langchain | 13358 | MIT |
| | yt-dlp/yt-dlp | 2701 | Unlicense |
| | tensorflow/models | 3632 | Apache 2.0 |
| | tiangolo/fastapi | 3056 | MIT |
| | keras-team/keras | 7310 | Apache 2.0 |
| | localstack/localstack | 5641 | Apache 2.0 |
| | geekan/MetaGPT | 773 | MIT |
| | 3b1b/manim | 782 | MIT |

Table 9: List of repositories and total number of PRs collected for four languages.

Table 10: Contrasting numbers of processed pull requests at the beginning of data collection and at the end as well as average repository size measured in number of files.

| Language | Total #repos collected | Total #PRs collected | Total #PRs w/ tests | Final # samples | Final # repos | Avg. repository size (files) |
|---|---|---|---|---|---|---|
| Java | 17 | 69 374 | 1433 | 165 | 6 | 2420.6 |
| JavaScript | 12 | 138 713 | 3136 | 1078 | 4 | 3706.5 |
| Python | 10 | 57 327 | 1012 | 199 | 6 | 1928.1 |
| TypeScript | 10 | 151 836 | 3042 | 729 | 5 | 8946.0 |

