# OpenReview forum: "SWE-PolyBench: A multi-language benchmark for repository level evaluation of coding agents"
_ICLR.cc/2026/Conference — Submitted to ICLR 2026_

### Official Review · Reviewer_mhYU · 2025-10-30

**Soundness:** 3
**Presentation:** 2
**Contribution:** 3
**Rating:** 4
**Confidence:** 3

**Summary:**

The paper introduces SWE-PolyBench, a 2,110-instance, four-language, repository-level benchmark plus a 382-instance human-curated subset, along with execution harnesses and CST-based retrieval metrics. The authors adapt Aider, SWE-Agent, and Agentless to the new multi-language setting and report performance stratified by language, task type, and syntactic complexity.

**Strengths:**

* Releases a comparatively large multi-language benchmark with automated execution harnesses, addressing the Python-centric focus of SWE-Bench and its verified variant.
* Provides a human-validated subset and per-instance metadata (task type, CST complexity, issue informativeness), enabling more nuanced analyses than prior datasets such as Multi-SWE-Bench.
* Introduces file- and CST-node-level retrieval metrics that complement pass rate.
* Documents the engineering required to adapt three open-source agent frameworks to multi-language settings—a contribution that will help practitioners reproduce baselines.

**Weaknesses:**

* The comparison to Multi-SWE-Bench remains qualitative. A quantitative head-to-head evaluation (size, language overlap, pass-rate baselines, metric coverage) is needed to clarify incremental novelty.
* All primary baselines share the same proprietary backbone (Claude 3.5 Sonnet). This makes it hard to disentangle dataset difficulty from access to a strong closed model; more open-weight or smaller-model results would increase reproducibility.
* The CST-based retrieval metrics are defined but only lightly interpreted; no validation (e.g., correlation with pass rate, human judgment) is given to show their diagnostic value.

**Questions:**

* Can you provide empirical comparisons with Multi-SWE-Bench beyond qualitative claims?
* Could you release additional baselines using fully open-weight models across all three agent pipelines to strengthen reproducibility?
* Do the CST retrieval metrics correlate with human notions of localization success or with downstream resolution rate? Any ablation studies would help establish their utility.

---

### Official Review · Reviewer_s8dR · 2025-10-31

**Soundness:** 2
**Presentation:** 2
**Contribution:** 2
**Rating:** 4
**Confidence:** 3

**Summary:**

This paper introduces SWE-PolyBench, a benchmark for evaluating LLM bug-fixing across multiple languages, including Python, Java, C++, JavaScript, and Go.
It is built automatically from GitHub repositories with bug–fix commit pairs, issue descriptions, and unit tests.
The benchmark offers a unified pipeline and standardized format for cross-language evaluation.
Experiments on several LLMs show performance gaps across languages, highlighting the challenge of code repair beyond Python.

**Strengths:**

1. The paper extends SWE-bench to a multilingual setting, with an automated data pipeline built from GitHub repositories containing buggy–fix commit pairs, issue descriptions, and unit tests.
2. It supports cross-model and cross-language comparison, providing a standardized framework for evaluating LLMs’ bug-fixing ability.
3. The paper is clearly written and presents useful experimental results.

**Weaknesses:**

1. The dataset includes several programming languages but is unevenly distributed, with some languages underrepresented.
2. This benchmark primarily builds on SWE-bench by extending the evaluation to four programming languages.Although it adds code localization assessment, the evaluation still centers on functional correctness. Maybe the methodological innovation remains limited.

**Questions:**

See weakness please

---

### Official Review · Reviewer_p2Ta · 2025-11-01

**Soundness:** 2
**Presentation:** 2
**Contribution:** 2
**Rating:** 2
**Confidence:** 5

**Summary:**

This paper introduces SWE-PolyBench, a new, large-scale benchmark designed to evaluate repository-level coding agents. The authors identify key limitations in existing benchmarks like SWE-Bench, namely their over-representation of the Python language, a focus on bug fixes, and the dominance of single repositories.
The authors evaluate three open-source agents (Aider, SWE-agent, Agentless) and find that current agents have uneven performance across languages (performing best on Python) and struggle significantly with complex, multi-file changes.

**Strengths:**

- The introduction of CST node-level retrieval metrics is a key innovation. It provides a formal way to measure an agent's ability to navigate a codebase and find the correct location for an edit, which is a critical sub-problem of the overall task. The analysis distinguishing retrieval success from pass-rate success is insightful.

- The decision to release both a large dataset (2110 instances) and a smaller, high-quality, human-annotated SWE-PolyBench_Verified (382 instances) is excellent.

- The paper provides an evaluation harness, Docker environments, and states that the code and dataset are included in the supplement, demonstrating a strong commitment to reproducibility.

**Weaknesses:**

- The research landscape is becoming saturated with benchmarks styled after SWE-Bench, such as SWE-bench Multilingual and Multi-SWE-bench. Many of these are iterative extensions focusing primarily on multi-language support. This trend raises concerns about diminishing returns. Simply introducing another 'same-style' benchmark, without offering a novel methodology or a substantive solution to the core challenges of agent evaluation, contributes little in terms of novelty to the field. Honestly, nearly repeated work is not proper for ICLR.

- LLM-based Classification: The paper (and its Appendix C ) notes that the classification of task types (e.g., "bug fix", "feature request") and the informativeness of issue descriptions (Figure 3)  were performed using a zero-shot LLM. While this is a practical approach for a dataset of this scale, it introduces a potential source of noise and bias. The authors commendably note this in the limitations , but it is a methodological weakness.

- Limited Agent Pool in Evaluation: The evaluation focuses on three open-source agents. While this provides a good baseline, the paper would have been even stronger if it included results from leading proprietary models to establish a state-of-the-art "upper bound" on this new, more difficult benchmark.

**Questions:**

- In your analysis, to what extent do you believe the agents' poor performance on Java, JS, and TS (Table 3) is a fundamental limitation of the LLMs' multilingual coding ability, versus a remaining artifact of the agent tooling (e.g., parsers, execution logic) being less mature for those languages?

---

### Official Review · Reviewer_d75x · 2025-11-01

**Soundness:** 3
**Presentation:** 3
**Contribution:** 2
**Rating:** 4
**Confidence:** 4

**Summary:**

This paper proposes SWE-PolyBench, a multi-language code repository-level benchmark that includes 2110 instances covering four languages: Java, JavaScript, TypeScript, and Python. The paper invests significant effort in data collection, experimental design, and engineering implementation, introducing innovative metrics such as syntax tree-based complexity analysis. However, overall, this is an incremental work, with the main contribution being the extension of the existing SWE-Bench approach to a multi-language scenario. The experimental results and conclusions are relatively expected, resembling more of a technical report rather than groundbreaking research.

**Strengths:**

1. Scientifically rigorous data collection process: Ensures data quality through metadata filtering (100+ PRs, recent updates, licenses, etc.) and runtime test validation (F2P testing).

2. Broad coverage and authenticity:

   * Spans 4 mainstream programming languages and 21 real open-source repositories.

   * Includes various task types (bug fixes, new features, refactoring, etc.).

   * Provides a high-quality verified subset (SWE-PolyBench Verified, 382 instances).

3. Innovative complexity measurement method: Node-level retrieval metrics based on concrete syntax trees (CST), enabling fine-grained evaluation of agents' ability to locate and modify code.

4. Substantial engineering investment:

   * Adaptation of multi-language Docker environments.

   * Modification of three open-source agents to support multiple languages.

   * Provision of a complete evaluation framework and reproducible code.

**Weaknesses:**

Inherent limitations of the dataset (some are acknowledged by the authors):

1. Limited task coverage: Focuses on "head-of-distribution" tasks, missing long-tail issues in daily software engineering.

2. Incomplete evaluation metrics:

   * Does not examine code quality dimensions (best practices, maintainability, security vulnerabilities, style consistency).

   * Test-based evaluation limits task types and penalizes reasonable variants (e.g., naming differences).

3. Lack of manual verification: Not all tasks have been confirmed by human experts to be solvable based on the given information; there may be instances with ambiguous or incomplete descriptions.

4. Reliability risks in LLM labeling: Task classification and information completeness assessment rely on zero-shot LLM, which may introduce biases.

5. Data contamination risks: Publicly available data may have been included in LLM training sets, affecting evaluation validity.

Deficiencies in experimental design:

6. Insufficient agent coverage: Only evaluates 3 open-source agents (Aider, SWE-agent, Agentless), lacking comparisons with commercial systems (e.g., GitHub Copilot, Cursor).

7. Single-dimensional CST analysis: Does not consider other complexity dimensions such as code length, dependency graph complexity, or cross-module calls.

8. Issues with experimental reproducibility:

   * Only reports single-run results, without providing variance analysis from multiple runs.

   * Does not discuss the impact of randomness on results.

9. Lack of ablation studies: Does not test the effects of known effective methods like test-time scaling or different prompt strategies in multi-language scenarios.

**Questions:**

1. **Necessity of manual verification:** Given that the quality of problem descriptions directly affects task solvability, are there plans to conduct manual expert verification on at least a portion of the instances? Particularly for tasks in the "insufficient description" category.

2. **Impact of language-specific context:** TypeScript's pass rate is significantly lower than other languages (5%-13%). If language-specific context (e.g., syntax rules, best practices, few-shot examples) is provided, would this gap narrow? Could this reveal whether the root cause is inherent language difficulty vs. uneven training data distribution?

3. "Retrieve-Generate" bottleneck analysis: Table 6 shows that Agentless-PB has the highest retrieval metrics on Python (File Recall 60.9%), but not the highest pass rate. Can the relative contributions of retrieval bottlenecks (failing to find the correct files) vs. generation bottlenecks (finding files but generating incorrect code) be quantified? Is this phenomenon more severe in other languages?

4. Mitigation measures for data leakage: The paper acknowledges data leakage risks but does not provide solutions. Have you considered adopting:

   * Time-based partitioning (e.g., only retaining PRs after 2024, later than mainstream LLM training cut-off dates).

   * Membership Inference Attacks to detect if the model has memorized training data.

   * Or other decontamination strategies?

5. **Effects of test-time scaling methods:** Various TTS enhancement methods on SWE-Bench (e.g., SE-Agent's [1] improvements to SWE-Agent) can bring significant gains. Are these methods equally effective on SWE-PolyBench? Is the improvement magnitude consistent across languages?

Among them, questions 1, 2, and 5 are more important. I am currently inclined to give a score of 6-8, and if my concerns can be addressed, I am very willing to raise the score.

---

References: [1] SE-Agent: Self-Evolution Trajectory Optimization in Multi-Step Reasoning with LLM-Based Agents

---

### Meta-Review · Area_Chair_dhER · 2025-12-27

**Summary:**

This paper proposes a new multilingual SWE-style benchmark. I have carefully read the paper and all reviews. I agree with reviewers that the proposed benchmark has some valuable data, but is not rigorously designed, and the overall novelty is not enough for ICLR, hence I recommend rejecting this paper.

**Reviewer Concerns:**

1. uneven distribution of the programming language: in the proposed dataset, different programming languages do not follow a balanced distribution. Hence, the results might not be informative enough.
2. As mentioned by multiple reviewers, the evaluation criteria are not well designed. For example, it does not examine code quality beyond pass rate.
3. The overall novelty is limited. The idea of repository-level SW evaluation is not novel, and the proposed dataset does not show unique value in terms of quality.

**Reviewer Scores:**

I do not think the reviewers will change their scores.

---

### Decision · Program_Chairs · 2026-01-26

Reject